# Proteobacteria and Firmicutes Secreted Factors Exert Distinct Effects on *Pseudomonas aeruginosa* Infection under Normoxia or Mild Hypoxia

**DOI:** 10.3390/metabo12050449

**Published:** 2022-05-17

**Authors:** Anna Charalambous, Evangelos Grivogiannis, Irene Dieronitou, Christina Michael, Laurence Rahme, Yiorgos Apidianakis

**Affiliations:** 1Department of Biological Sciences, University of Cyprus, Nicosia 2109, Cyprus; annita@ucy.ac.cy (A.C.); egrivogiannis@gmail.com (E.G.); irened@cytanet.com.cy (I.D.); michael.m.christina1@ucy.ac.cy (C.M.); 2Department of Surgery, Massachusetts General Hospital and Harvard Medical School, Boston, MA 02115, USA; rahme@molbio.mgh.harvard.edu

**Keywords:** acetic acid, lactic acid, midgut, mucosa, peptidoglycan, quorum sensing, sugar fermentation

## Abstract

Microbiota may alter a pathogen’s virulence potential at polymicrobial infection sites. Here, we developed a multi-modal *Drosophila* assay, amenable to the assessment of human bacterial interactions using fly survival or midgut regeneration as a readout, under normoxia or mild hypoxia. Deploying a matrix of 12 by 33 one-to-one *Drosophila* co-infections via feeding, we classified bacterial interactions as neutral, synergistic, or antagonistic, based on fly survival. Twenty six percent of these interactions were antagonistic, mainly occurring between Proteobacteria. Specifically, *Pseudomonas aeruginosa* infection was antagonized by various *Klebsiella* strains, *Acinetobacter baumannii*, and *Escherichia coli*. We validated these interactions in a second screen of 7 by 34 one-to-one *Drosophila* co-infections based on assessments of midgut regeneration, and in bacterial co-culture test tube assays, where antagonistic interactions depended on secreted factors produced upon high sugar availability. Moreover, *Enterococci* interacted synergistically with *P. aeruginosa* in flies and in test tubes, enhancing the virulence and pyocyanin production by *P. aeruginosa*. However, neither lactic acid bacteria nor their severely hypoxic culture supernatants provided a survival benefit upon *P. aeruginosa* infection of flies or mice, respectively. We propose that at normoxic or mildly hypoxic sites, Firmicutes may exacerbate, whereas Proteobacteria secreted factors may ameliorate, *P. aeruginosa* infections.

## 1. Introduction

The persistent use of antibiotics leads to multi-drug resistant bacteria [1,2,3,4]. The so-called “ESKAPE” pathogens, including *Enterococcus faecium*, *Staphylococcus aureus*, *Klebsiella pneumoniae*, *Acinetobacter baumannii*, *Pseudomonas aeruginosa*, and *Enterobacter* sp., more often than others escape the bactericidal action of antimicrobial agents and are responsible for the majority of hospital-acquired poly-microbial infections [1,5]. Their modes of action include drug inactivation by enzymatic cleavage, the modification of the bacterial target (the antibiotic binding site), and reduced drug accumulation inside the bacteria, either due to reduced permeability or by increased drug efflux [6]. The ESKAPE pathogens also form biofilms that physically prevent the host cells from mounting immune responses and antibiotics from inhibiting pathogens. Biofilms protect specialized dormant persister cells, tolerant to antibiotics, that cause difficult-to-treat infections [7]. Given the drastically declining number of effective antibiotics, it is imperative to explore novel approaches for preventing and treating poly-microbial infections, including detailed studies of microbial species interactions, using appropriate infection models.

*Drosophila* bears significant anatomical, immunological, and physiological similarities to humans and entails an easy and inexpensive infection host [8,9]. Thus, it may serve as a suitable screening tool prior to using the more complex mammalian infection models [10]. Although the fly does not have an adaptive immune system as we know it in humans, it mounts a robust innate immune response, resembling its mammalian counterparts [11,12]. Flies deploy plasmatocytes as phagocytes and signal transduction pathways in various tissues, leading to the systemic and localized production of anti-microbial peptides [13]. Moreover, midgut regeneration [14] and external epithelial wound closure and melanization re-establish the barrier against foreign elements [15,16,17,18].

Different conditions, including oxygen and nutrient availability, temperature, pH, and disease state influence microbial community composition at various anatomical sites and consequently affect microbial species interactions and metabolites produced [19]. More specifically, limited oxygen availability in the skin, lungs, and gut mucosa creates a mildly hypoxic environment that contributes to microbiome diversity [19,20,21]. This diversity in turn facilitates a plethora of microbial interactions, some of which have been described in several models, including a murine chronic wound model, a rat lung-infection model, and a *Drosophila* infection model, with the mechanism of action fully delineated [22,23,24,25,26]. These primarily include host-detrimental synergistic interactions such as those occurring between *P. aeruginosa* and Gram positive bacteria (e.g., *S. aureus*) and involve *P. aeruginosa* sensing of Gram positive bacteria peptidoglycan that leads to a concomitant increase in quorum-sensing-regulated virulence factors, which modify bacterial community composition and enhance host killing [27]. In addition, host-protective antagonistic interactions have also been described for lactic acid-producing bacteria (e.g., *Bifidobacteria* and *Lactobacilli*) in the human gut [28,29]. However, oxygen availability is not the sole determining factor and many microbial interactions are at play in complex poly-microbial sites. For example, *P. aeruginosa* mouse gut infection can be inhibited by *Escherichia coli* in the presence of dietary vegetable fats, rather than carbohydrates or proteins, which favor *E. coli* fermentation and the production of lactic and acetic acid [30].

In this work, we have developed a *Drosophila* multi-modal pathology assay that facilitates the investigation of the impact of human bacterial interactions on fly survival and midgut regeneration, in a quantifiable way, under normoxia or mild hypoxia. Based on fly survival, 26% of the examined bacterial interactions were antagonistic, occurring mostly between species of Proteobacteria; ~9% were synergistic, and the rest were neutral. Notably, under both normoxic and mildly hypoxic conditions and based on both fly survival and midgut regeneration readouts, *P. aerugin**osa* displayed antagonistic interactions with *Klebsiella aerogenes, K. pneumoniae*, and *E. coli*. In contrast, *P. aeruginosa* displayed synergistic interactions with bacteria belonging to the phylum Firmicutes. In addition, bacterial co-culture test-tube assays showed that secreted factors rather than direct bacterial contact mediated inhibition and that [glucose]_high_ in the media was necessary. Moreover, co-infection of flies with *P. aeruginosa* and lactic acid bacteria did not display any benefit on survival in mice since supernatants from severely hypoxic bacterial cultures of *Lactobacillus plantarum* and *Bifidobacterium infantis* failed to ameliorate *P. aeruginosa* mouse lung infection, as opposed to supernatants of *A. baumannii* and, potentially, *Klebsiella* species.

## 2. Results

### 2.1. An Array of Culturable Strains of Bacteria Found in Humans Can Stably Associate with Drosophila via Feeding

To investigate the outcome of interactions between culturable human bacterial pathogens, we established a multi-modal *Drosophila* assay that enables fast, accurate, and reproducible results, in a quantifiable manner, with fly survival and midgut regeneration as readouts, under normoxia or mild hypoxia. Firstly, we evaluated the presence of various culturable human bacterial strains in *Drosophila* infected via feeding for up to five days. We determined that the stable association of examined bacteria with flies occurred when mated, gut-bacteria-cleaned females were used and then reared at 29 °C on infection media with 10% LB in 4% sucrose as a vehicle. Furthermore, as shown in Figure 1A, Colony Forming Unit (CFU) measurements revealed that infected flies harbored between 10^3^ to 10^8^ bacteria per fly, depending on the bacterial strain, at day one of feeding. Bacterial presence was maintained throughout the five days for all strains tested; most sustained a constant level (<0.5 log_10_ increase or decrease in bacterial CFUs by day 5), including *Vibrio cholerae*, *Yersinia enterocolitica*, *Serratia marcescens*, *Morganella* sp., *K. aerogenes, P. aeruginosa* (PA14), *K. pneumoniae, E. faecalis, Lactobacillus acidophilus, L. plantarum*, and *B. Infantis*, while *Salmonella paratyphi, Salmonella dupentery*, and *Lactobacillus brevis* showed an increase in bacterial CFUs by day 5 (>0.5 log_10_ increase in bacterial CFUs by day 5). Some showed a decrease of 0.5–1.0 log_10_ in bacterial CFUs by day 5, including *E. coli* (EPEC-O127:H6 E2348/69 and DH5α), *Citrobacter* sp., *Enterococcus* sp*.*, and *Listeria monocytogenes*, while others showed a greater decrease (>1 log_10_ in bacterial CFUs), including *Proteus* sp., *Proteus mirabilis, Enterobacter cloacae, A. baumannii, Yersinia pseudotuberculosis, Salmonella choleraesuis, Salmonella typhi, Klebsiella* sp., *Providencia* sp., *Pasteurella multocida*, and *Streptococcus agalactiae* (Figure 1A).

Moreover, we assessed the virulence of the same strains by evaluating the fly survival rate, i.e., the time in days it takes for 50% of the flies to die (LT50) upon infection via feeding. As shown in Figure 1B, Proteobacteria vary in virulence, with some having detrimental effects on fly survival, most notably *P. aeruginosa* (PA14) (LT50 = 4 days), *Yersinia tuberculosis* (LT50 = 9 days), *Klebsiella* sp. (LT50 = 9 days), *S. marcescens* (LT50 = 11 days), *A. baumannii* (LT50 = 14 days), *P. mirabilis* (LT50 = 14 days), *Proteus* sp. (LT50 = 15 days), *Morganella* sp. (LT50 = 16 days), *S. choleraesuis* (LT50 = 16 days), *P. multocida* (LT50 = 17 days), and *K. aerogenes* (LT50 = 18 days). In contrast, the Gram positive bacteria examined, including 5 strains of lactic acid bacteria, were not pathogenic to the flies (LT50 was not reached 20 days post infection), with the exception of *Enterococcus* sp. (LT50 = 13 days). These results suggest a correlation between the pathogenicity observed in the *Drosophila* and that encountered in humans. Slight deviations observed can be accounted for by a number of factors, including the diversity of resident microbiome at the infection site. We, therefore, considered the *Drosophila*, under the conditions established here, to be a suitable model to examine interactions between culturable bacterial strains found in humans as these may occur in poly-microbial normoxic or mildly hypoxic settings that allow for fast, accurate, and reproducible results, in a quantifiable manner (Figure 1B).

### 2.2. Graded Classification of Bacterial Interactions as Neutral, Synergistic, or Antagonistic, Based on Drosophila Survival Rate upon Feeding Infection

To identify bacterial strains that alleviate the virulence/pathogenicity of highly virulent bacterial strains in *Drosophila*, we performed a 12 by 33 one-to-one bacterial combination screen of fly survival rate following infection upon feeding. We evaluated the LT50 upon infection with each of the 12 strains identified as highly virulent, based on preliminarily assessments having fly survival and midgut regeneration as readouts. We tested these strains either alone (LT50_H_) or in combination with each of the remaining bacterial strains (LT50_H+X_). For every one-to-one strain combination tested, the highest virulence strain was taken as a reference, while the other was taken as a potential inhibitor of the first. Figure 2 indicates that most of the interactions tested were neutral (white boxes), that is, they had neither a positive nor a negative effect. Twenty eight out of the 318 combinations tested (8.8%) exhibited synergistic effects, accelerating the time of fly mortality either drastically by more than 50% (LT50_H_/LT50_H+X_ > 1.5) or mildly between 20 and 50% (LT50_H_/LT50_H+X_ = 1.2–1.5), indicated with red and pink boxes, respectively, in Figure 2. These were primarily but not exclusively interactions between the 12 highly virulent bacterial strains. Paradigms of strong synergism included (a) *Y. pseudotuberculosis* with *P. multocida*, (b) *S. marcescens* with *S. dupentery* or *K. aerogenes*, (c) *P. multocida* with *P. mirabilis*, and (d) *Morganella sp.* or *S. dupentery* with *Klebsiella* sp. Paradigms of mild synergisms included: (a) *Y. pseudotuberculosis* with *S. marcescens* or *Klebsiella* sp*.*; (b) *S. marcescens* with *Morganella* sp*.*, or *Klebsiella* sp. or *A. baumannii*; (c) *P. multocida* with *Klebsiella* sp.; (d) *Morganella* sp. with *S. dupentery*; (e) *S. dupentery* with *Citrobacter* sp.; and (f) *K. aerogenes* with *A. baumannii* or *Citrobacter* sp.

Strong synergistic interactions were prominently observed between the highly virulent *P. aeruginosa* (PA14) and Firmicutes species, in accordance with previous studies [26,31]. Specifically, fly mortality timing was significantly accelerated when flies were co-infected with *P. aeruginosa* and *Enterococcus* sp. or *E. faecalis*, dying on average at 2 days of feeding infection, compared to more than 4 days following infection with *P. aeruginosa* alone (Figure 3A) [31]. Moreover, co-infection with the probiotic lactic acid bacteria, *L. acidophilus* and *L. plantarum*, did not benefit the host against *P. aeruginosa* infection but instead tentatively increased fly mortality by 20–50% (LT50_H_/LT50_H+X_ = 1.2–1.5; Figure 3B). The latter observation may depend on our experimental setup, which involved simultaneous administration of equal numbers of bacteria so they could interact on equal terms without one having a numerical or timing advantage over the other. Under other conditions lactic acid bacteria may establish host defense prior to an infection.

Strikingly, we identified 83 antagonistic interactions or 26% out of the 318 combinations tested. These were either strong (LT50_H_/LT50_H+X_ < 0,5), leading to a more than 50% reduction in the timing of fly mortality, or mild (LT50_H_/LT50_H+X_ = 0.5–0.8). Seven out of the 10 strong antagonistic interactions observed involved *P. aeruginosa* (dark green boxes in Figure 2). Sixteen bacteria strains in total, spanning the whole spectrum of virulence, antagonized infection with *P. aeruginosa* (PA14): 7 of them strongly (*S. marcescens, K. aerogenes, Klebsiella* sp., *A. baumannii, S. choleraesuis, S. paratyphi*, and *Providencia* sp.) (dark green boxes in Figure 2), and the following 9 mildly: *Proteus sp*., *Citrobacter* sp., *E. cloacae, E.coli (EPEC), K. pneumoniae, S. typhi, S. agalactiae, L. paralimentarius*, and *Y. enterocolitica* (light green boxes in Figure 2).

Infection with *Y. pseudotuberculosis* was also mildly antagonized by 9 strains, including *S. dupentery, Citrobacter* sp., *E.coli* (EPEC and DH5α), *S. paratyphi, V. cholerae, Y. enterocolitica, L. monocytogenes*, and *Providencia* sp., while infection with *S. marcescens* was mildly antagonized by 13 strains, including *P. mirabilis, E. cloacae, E.coli* (EPEC and DH5α), *S. agalactiae, Enterococcus* sp., *E. faecalis, K. pneumoniae, S. paratyphi, V. cholerae, Y. enterocolitica, L. monocytogenes*, and *Providencia* sp.

Infection with *P. multocida* was strongly antagonized by *Y. enterocolitica* and *Morganella* sp. and mildly by 12 strains, while infection with *K. aerogenes* was antagonized by *E. cloacae* and 9 other strains. Infection with either *Proteus* sp., *P. mirabilis*, or *Morganella* sp. was mildly antagonized by 4 strains, and infection with *S. dupentery* was mildly antagonized by 3 strains. In contrast, infection with *Citrobacter sp.* was moderately antagonized by *E. cloacae* and mildly antagonized by 2 strains. Co-infection with either *Klebsiella* sp. or *A. baumannii* did not exhibit synergistic interactions (Figure 2).

In summary, most interactions between highly pathogenic bacterial strains and other bacterial strains of varying host killing potential were neutral. However, 8.8% were synergistic and 26% were antagonistic, graded according to their impact on fly survival as strong or mild.

### 2.3. Classification of Interactions between Pathogenic Bacteria as Neutral, Synergistic, or Antagonistic, Based on Drosophila Midgut Cell Mitosis upon Feeding Infection

To preserve the integrity of the barrier epithelia of the mucosa, damaged cells are replenished via stem cell division [32]. The gut epithelium of *Drosophila* and mammals contains differentiated epithelial cells and progenitor cells constantly undergoing cell death, shedding, and division. Epithelial cells that are damaged or stressed by microbial and other exogenous or endogenous factors emit stem-cell-inducing cytokines and growth factors before their exfoliation, thus inducing their replenishment [33,34]. This regenerative inflammatory process is a host defense mechanism protecting the epithelium from bacterial attachment and colonization [35]. It helps to expel colonized pathogens, confine bacterial spreading, and restrain excessive inflammation [36]. We thus sought to determine the extent of *Drosophila* midgut stem cell division in response to combinatorial bacterial infections by enumerating midgut mitosis.

The infection of flies with lactic acid bacteria, including *B. infantis, L. paralimentarius, L. plantarum, L. acidophilus*, and *L. brevis*, and bacterial strains known to exhibit reduced virulence such as *E. coli* (DH5a) had a minimal effect on epithelial cell turnover (Figure 4A, framed in green rectangles) [14,37,38,39]. On the other hand, known pathogens, including *A. baumannii, Klebsiella* sp., *L. monocytogenes, Enterococcus* sp. (and *E. cloacae*), *Proteus* sp. (and *P. mirabilis*), *Morganella* sp., *P. multocida, V. cholerae*, and *Bacteroides thetaiotaomicron*, had a higher impact on epithelial cell turnover (framed in an orange rectangle in Figure 4A). Additional bacterial strains known to be pathogenic to humans and flies, including *Citrobacter, P. aeruginosa* (PA14), *E. faecalis, B. fragilis, S. marcescens*, and *S. agalactiae*, as well as *E. coli* strains BWH and MGH, induced the highest intestinal stem cell divisions in a primary screen (framed in a red rectangle in Figure 4A). In a follow-up co-infection screen (Figure 4C), we assessed the 7 most mitosis-inducing strains against all 34 (potentially antagonistic) bacterial strains, shown in Figure 4A. Representative images of midgut mitosis upon co-infection of flies with *P. aeruginosa* (PA14) and *E. coli* are shown in Appendix A. Prominent antagonistic interactions were noticed between (a) *Citrobacter* sp. and *Proteus* sp.; (b) *P. aeruginosa* (PA14) and *V. cholerae* or *B. fragilis*; (c) *E. faecalis* and *V. cholerae, B. fragilis*, or *S. agalactiae*; (d) *E. faecalis* and *V. cholerae* or *B. fragilis*; (e) *S. agalactiae* and *P. aeruginosa (PA14*) or *B. thetaiotaomicron*; (f) *B. fragilis* and *S. agalactiae* or *Proteus* sp.; and (g) *S. marcescens* with *E. coli* or *E. faecalis* (Figure 4B,C).

The reduction in the number of mitotic cells was also observed when lactic acid bacteria, such as *B. infantis, L. plantarum, L. paralimentarium*, and *L. acidophilus*, interacted with highly virulent pathogens, such as *Citrobacter, P. aeruginosa* (PA14), *B. fragilis, E. faecalis*, and *S. agalactiae* (Figure 4B). Of note, the inhibition of increased mitosis upon infection did not closely correlate with extended fly survival described in the previous section, except in seven distinct cases of antagonistic interaction, namely, *E. coli* (MGH), *K. pneumoniae, S. agalactiae*, and *K. aerogenes*, inhibiting *P. aeruginosa* (PA14) virulence (Figure 4D), as well as *E. coli*, *E. faecalis*, and *V. cholerae*, inhibiting *S. marcescens* virulence (data not shown). We conclude that localized and systemic bacterial interactions are complex because intestinal damage and concomitant regeneration do not equal systemic toxicity and survival. However, some bacterial interactions can exhibit both local, intestinal, and systemic effects.

### 2.4. Secreted Factors from K. aerogenes, Klebsiella *sp.*, and A. baumanni Inhibit P. aeruginosa Growth and Pyocyanin Production in Aerobic Liquid Cultures Supplemented with Sugars

Cell-free supernatants from bacterial cultures are rich in factors that affect the growth of various bacteria [40,41,42,43,44,45,46]. Depending on the bacterial strain and the culturing conditions, biologically active substances can be secreted in bacterial supernatants, including bacteriocins [47]; metabolites, such as organic acids [48]; hydrogen peroxide [49]; biosurfactants [50]; and other immunomodulatory substances [51,52,53].

Changes in carbohydrate levels affect sleep, locomotion, longevity, and immunity in *Drosophila* [54,55,56,57]. To provide a minimal carbon source essential for fly survival, *Drosophila* infection media were supplemented with sucrose. The presence of sucrose though may provide additional benefits by facilitating bacterial interactions in co-infected flies. In accordance with our previous findings, that simple sugars (glucose or sucrose) are essential for *E. coli* strains to produce lactic and acetic acid and inhibit *P. aeruginosa* (PA14) growth [30], we now show that glucose is necessary to facilitate the inhibition of PA14 growth by the supernatants of Proteobacteria strains in liquid cultures (Figure 5A). We assessed PA14 bacterial growth in the presence of supernatants or heat-killed bacteria derived from *K. aerogenes, Klebsiella* sp., *A. baumannii, S. paratyphi, S. marcescens*, and *Providencia* sp., in aerobic LB cultures. Supernatants from *K. aerogenes* and *A. baumannii* significantly reduced PA14 CFUs by 7.8 and 4.5 log_10_, respectively. In contrast, supernatant from *Klebsiella* sp. reduced PA14 CFUs by 4.2 log_10_ but not statistically significantly. The reduction in bacterial growth correlated with the reduction in PA14-produced pyocyanin, a redox-active secondary metabolite and potent anti-bacterial (Figure 5B). In contrast, supernatants from *E. faecalis* and *Enterococcus* sp., shown above to interact synergistically with *P. aeruginosa* (PA14), led to tentatively increased pyocyanin production by PA14 (Appendix A). No inhibition of PA14 growth or pyocyanin production was observed when heat-killed bacteria (cell fractions) were used, even in the presence of glucose (Figure 4C,D). Therefore, we suggest that not only *E. coli* strains [30] but also other Proteobacteria, such as *Klebsiella* species and *A. baumannii*, ferment sugars into lactic and acetic acid aerobically, inhibiting *P. aeruginosa* (PA14) growth.

### 2.5. Antagonistic Interactions between P. aeruginosa and Selected Proteobacteria but Not Lactic Acid Bacteria Also Occur under Mild Hypoxia

Partial oxygen pressure (pO_2_) drops steeply along the radial axis from the intestinal submucosa to the lumen, displaying pO_2_ = 59 mm Hg (8%) in the small intestinal wall, around 22 mm Hg (3%) at the villus tip, and less than 10 mm Hg (2%) in the small intestinal lumen [21]. This hypoxic environment is initiated and maintained by a large and diverse microbial population residing in the human gut. Tissue oxygenation at the epithelia-lumen interface leads to an increase in the population of aerotolerant microbes. At the same time, the central portion of the lumen is rendered deficient in oxygen and is home to trillions of anaerobic microbes [58]. Thus, the fermentation process towards lactic acid production may be more efficient at the lumen or the mucosa, depending on the bacterial species found at each site [59].

To model the mildly hypoxic environment of the mucosa, we reared flies under hypoxic ([O_2_] = 5–5.5%) conditions that are tolerable for fly survival [60]. As shown in Figure 6A, flies infected via feeding under these hypoxic conditions with *P. aeruginosa* in combination with *Klebsiella* sp., *K. aerogenes*, or *E. coli* (MGH) exhibited a significant delay in the timing of fly mortality (*** *p* < 0.0001). Similarly, as shown in Figure 6B, co-infection with *P. aeruginosa* and *A. baumannii, S. paratyphi*, or *Providencia* sp. exhibited a moderate but significant delay (** *p* < 0.004). In contrast, co-infection with *P. aeruginosa* and lactic acid bacteria (*L. plantarum* or *B. infantis*) did not significantly affect fly survival, compared to infection with *P. aeruginosa* alone (Figure 6C), in this experimental setup.

### 2.6. Lactic Acid Bacteria Supernatants Produced under Severe Hypoxia Fail to Inhibit P. aeruginosa Virulence in a Mouse Lung-Infection Model

Lactic acid bacteria, including *L. plantarum* and *B. infantis*, thrive in severely hypoxic and anoxic conditions [61]. We therefore sought to examine whether lactic acid bacteria may confer protection against *P. aeruginosa* infection when grown under their optimal (anoxic) conditions. In addition, we wanted to examine *P. aeruginosa* inhibition by other bacteria in a mammalian setting. We used a mouse lung-infection model, infecting CD-1 female mice intranasally either with PA14 using either 4% sucrose in LB as a vehicle (control) or supernatants from various Gram negative or lactic acid bacteria, the latter derived from glucose-supplemented, anoxic cultures. As shown in Figure 7, when PA14 was administered in the presence of *A. baumannii* supernatant as a vehicle, mouse survival at 48 h post-infection was 60%, compared to 5% of the control. Similarly, infection with PA14 in the presence of *Klebsiella* sp. or *Providencia* sp. culture supernatants had a tentative impact on mouse survival (40% survival by 48 h), which, however, was not statistically significant. However, *B. infantis* and *L. plantarum* failed to confer any noticeable or significant delay in mouse mortality (Figure 7). Thus, factors secreted by lactic acid bacteria, even when produced under anoxia, are of limited potency in an infection setting.

## 3. Discussion

Clinically important infections are frequently polymicrobial; thus, the delineation of microbe–microbe, as well as host–microbe, interactions can provide a better understanding of infection establishment and evolution upon treatment [62,63]. Accordingly, we developed the multi-modal *Drosophila* assay, which provides several advantages. Firstly, it is amenable to the assessment of pathogenic bacterial interactions in two biological processes, namely, fly survival and midgut regeneration. Secondly, even though only one-to-one bacterial interactions were investigated in this work, this assay may easily be adapted to investigate more elaborate, polymicrobial interactions, representative of microbial populations at specific human anatomical sites that are prone to infection. In addition, this assay can be specifically tailored according to factors and conditions present in different mucosal sites, such as low pH and low nutrient and oxygen availability, which may affect the microbes’ ability to synergize or antagonize each other. Anoxic interactions are seemingly prohibitive, given that flies are small, and ambient oxygen can always reach the gut lumen. However, the skin, the lungs, the vagina, and the intestinal mucosa are not anoxic; moreover, the contribution of obligate anaerobes, found in the anoxic gut lumen, can also be studied in this assay by feeding flies with the nutrients and metabolites these bacteria may produce in culture [62,63].

Pathogenic bacteria may act synergistically or antagonistically in inducing virulence traits, altering the infected niche, forming biofilms, or modulating the host immune response [64,65,66] via cell-to-cell signaling, metabolite exchange, or cell–cell contact [67,68]. Even though bacterial antagonism was first described over a century ago [69], it has been studied less extensively. As a result, specific examples of bacterial strains resisting pathogen colonization and thus protecting the host against infection comprise a short list, which includes (a) *Clostridium scindens* inhibiting *C. difficile* infection, (b) non-toxigenic *Bacteroides fragilis* resisting infection by the enterotoxic *B. fragilis*, (c) *E. coli* O21:H+ inhibiting muscle atrophy, and (d) *E. coli* ECN acting against intestinal pathogens [70]. The work presented here is, to our knowledge, the first systematic approach to examine an extensive array of interactions between culturable human bacteria providing an alternative model to reveal novel antagonistic interactions between human pathogenic bacteria.

The one-to-one interactions between human bacterial stains studied in the multi-modal *Drosophila* assay we established revealed that synergistic interactions occurred primarily between 12 highly virulent bacterial strains. Strong synergism, however, was also observed between the highly virulent *P. aeruginosa* clinical strain PA14 and bacteria of the Firmicutes phylum, such as *Enterococcus* sp. These synergistic interactions could be explained by the previously reported sensing of Gram positive bacteria peptidoglycan and the concomitant increase in virulence factor production by *P. aeruginosa* [31]. The interaction between *P. aeruginosa* and *Enterococcus* sp. was recently shown to facilitate the thickness of a two-species biofilm and adaptation to harsh environments, such as resistance to antibiotics [71,72,73].

Most importantly, however, the *Drosophila* assay provides evidence of 85 clinically relevant, previously unrecognized antagonistic bacterial interactions. Prominent among those is the inhibition of *P. aeruginosa* by Proteobacteria, three of which, *K. aerogenes*, *K. pneumoniae*, and *E. coli*, also affected both fly survival and midgut cell mitosis. Additional Gram negative bacterial strains antagonized *P. aeruginosa* infection, reducing the number of dividing intestinal stem cells in the fly midgut, but *Lactobacilli* did not exhibit such an antagonistic effect. This might be because of the multi-modal *Drosophila* assay we developed here, which is aimed at a quantifiable assessment of interactions between equal numbers of culturable human bacteria upon simultaneous co-infection, without any one of the bacterial strains having a priori a numerical or timing advantage over the other. We acknowledge that this might differ from other settings where *Lactobacilli* are resident at specific sites, namely, the gut, and therefore are most likely to outnumber pathogenic bacteria upon infection, enabling host-protective actions. In addition, given the mild hypoxia under which the flies were reared, the lactic acid bacteria tested might not ferment sugars effectively, and the shedding of peptidoglycan from these bacteria might be tilting the balance towards the stimulation of *P. aeruginosa’s* virulence rather than inhibition. On the other hand, *P. aeruginosa* inhibition might be preferentially mediated under hypoxic conditions by certain Proteobacteria prominently found in the human mucosa. In addition, in this specific experimental setting, we only observe marginal differences in epithelial cell mitosis between flies infected with *Lactobacilli* alone and control flies. This may seem to contrast a previous report showing that *Lactobacilli* can induce ROS-dependent cellular proliferation in the *Drosophila* intestine [74]. We hypothesize that this might be due to the specific *Lactobacillus* strains, the fly food recipe used, or the prolonged (48-h) interaction of adult flies with the bacteria.

Using test tube co-culture assays (under aerobic conditions), we evaluated *P. aeruginosa* growth and pyocyanin production, in the presence or absence of glucose. Pyocyanin can accept electrons from NADH generated during *P. aeruginosa* carbon source oxidation, producing oxygen radicals [75]. It is the most prominent quorum sensing metabolite, facilitating *P. aeruginosa* fitness and biofilm formation and antibiotic resistance [76,77,78]. In our experiments, glucose facilitated *K. aerogenes*, *K. pneumoniae*, and *A. baumannii* to produce secreted factors that inhibit the growth of *P. aeruginosa* and its ability to secrete pyocyanin. Moreover, a third *Klebsiella* strain (*Klebsiella* sp.) was identified in our screen of fly survival as an antagonist of *P. aeruginosa*. Thus, species of the genus *Klebsiella* appear to bear prominent features, presumably a high ability to ferment sugars into lactic or acetic acid in the presence of oxygen. Given that *P. aeruginosa* often coexists with *K. pneumoniae* upon bacterial pneumonia in the human lungs [79], the interaction between these species warrants further analysis. Interestingly, the antagonistic interactions observed under normoxia hold true even under mild hypoxic conditions, similar to those observed in the mammalian gut submucosa and lung mucosa. We aimed to mimic the hypoxic gut mucosa and the mildly hypoxic lung mucosa by rearing flies at [O_2_] = 5.5%, and by using the mouse intranasal lung infection model. We found that Proteobacteria prominently antagonized *P. aeruginosa* infection under mild hypoxia, in contrast to lactic acid bacteria. More specifically, *K. aerogenes*, *K. pneumoniae*, and *E. coli* significantly prolonged fly survival upon *P. aeruginosa* infection, but so did *S. paratyphi, Providencia* sp*.*, and *A. baumannii*, even though less significantly. However, *B. infantis* and *L. plantarum* failed to confer significant inhibition of *P. aeruginosa* infection, even when reared under severe hypoxia, that is, the optimal conditions for their survival and growth.

We therefore propose that under normoxia or mild hypoxia ([O_2_] > 5%), which may occur in the mammalian skin, and the lung and gut mucosa, Firmicutes strains belonging to *Enterococci* and lactic acid bacteria do not undergo sugar fermentation to lactic and acetic acid ([lactic/acetic acid]_low_), as *E. coli* prominently does [30], and may instead shed their peptidoglycan ([peptidoglycan]_high_), which induces *P. aeruginosa* virulence [31]. In contrast, Proteobacteria, such as *E. coli* and *Klebsiella* species, may undergo sugar fermentation to lactic and acetic acid ([lactic/acetic acid]_high_), while their peptidoglycan is lowly stimulating ([peptidoglycan]_low_), enabling them to antagonize *P. aeruginosa* growth at the infection site (Figure 8).

This model could potentially provide an explanation as to why skin and open wounds that are mostly inhabited by Gram positive bacteria (*Streptococcus epidermidis* and *S. aureus*) are vulnerable to *P. aeruginosa* infection, whereas the small intestine and colon, which are rich in *Proteobacteria*, especially those of the family of *Enterobacteriaceae* that includes the genera of *Enterobacter, Escherichia, Klebsiella*, and *Salmonella*, are resistant to *P. aeruginosa* infection. The case of human lungs is more complicated as it is not clear whether the Proteobacteria identified within this study coexist with *P. aeruginosa* in diseased human lungs or if they appear at a later stage, in the same patients [79]. Knowing this might provide new opportunities to keep *P. aeruginosa* in check at this anatomical site.

## 4. Materials and Methods

### 4.1. Chemicals and Reagents

#### 4.1.1. LB/BHI Broth Media Preparation

LB (Candalab—Madrid, Spain) and BHI (brain heart infusion) broth (HIMEDIA—Mumbai/India) were prepared according to manufacturer instructions: LB (20 g/1 L ddH_2_O) and BHI (37 g/1 L ddH_2_O) were mixed, autoclaved, and stored at room temperature.

#### 4.1.2. LB Agar Plates/LB Agar + Rifampicin Plates Preparation

LB agar (Invitrogen—Waltham, MA, USA) plates were prepared according to manufacturer instructions: LB agar (32 g/1 L ddH_2_O) was autoclaved and poured in petri dishes. LB agar + 1mM Rifampicin (Rif) plates were prepared and stored at 4 °C. These were used in test-tube culture experiments only, to selectively measure the CFUs of the Rif-resistant strain PA14.

### 4.2. Bacterial Strains

Gram negative bacteria used in the experiments included *P. aeruginosa* (PA14), *Yersinia* (*pseudotuberculosis* and *enterocolitica*), *Serratia marcescens, Pasteurella multocida, Klebsiella* (*aerogenes, pneumoniae* and sp.), *Acinetobacter baumannii, Providencia* sp., *Proteus* (*mirabilis* and *sp*.), *Morganella* sp., *Salmonella* (*dupentery, choleraesuis, paratyphi*, and *typhi*), *Citrobacter* sp., *Enterobacter cloacae, Enteropathogenic E. coli* (EPEC O127:H6 E2348/69), *E. coli* strains DH5α, BWH (human isolate from BWH hospital, Boston) and MGH (uropathogenic human isolate from MGH hospital, Boston), *Vibrio cholerae*, and *Bacteroides* (*fragilis* and *thetaiotaomicron*). Gram positive bacteria used in the experiments included *Streptococcus agalactiae, Listeria monocytogenes, Enterococcus* (*faecalis* and sp.), *Bifidobacterium infantis*, and *Lactobacillus* (*brevis, acidophilus, plantarum*, and *paralimentarius*). *P. aeruginosa* strains were previously described [80]. *E. coli* strains were obtained from Prof. Elizabeth Hohmann at Mass General Hospital, Boston, USA [30]. *L. plantarum* and *L. brevis* were obtained from James Angus Chandler [81]. The rest of the strains were obtained from Lynn Bry, MD, PhD at the Microbiology Laboratory, Brigham and Women’s Hospital, Boston, USA. All strains were maintained as glycerol stocks and stored at −80 °C.

### 4.3. Drosophila Diet, Maintenance, and Experiments

Oregon R flies were used as a standard wild-type strain, reared on a standard, endotoxin-free, agar/cornmeal diet (1% Agar, 3% Yeast, 5% Sugar, and 6% Cornmeal, supplemented with 2.56% Tegosept and 0.38% Propionic Acid) and kept in plastic bottles, with approximately 50 mL of fresh fly food in a 12-h day and night cycle, at 25 °C with 65% humidity. For maintenance, flies were transferred to new bottles with fresh fly food every 3 to 4 days. For fly infection experiments, newly hatched flies were transferred for 4 days to vials containing fresh fly food with preservatives (propionic acid and Tegosept), to practically eliminate culturable microbiota and avoid pre-treatment with antibiotics that may interfere with subsequent bacterial infection and colonization.

#### 4.3.1. Fly infection and Survival

This is an adaptation of a previously described protocol [14]. Oregon R female flies 4–7 days old, previously starved for 5–6 h, were fed with an infection mix containing one or two bacterial strains at a time. Survival was assessed every day based on the number of dead and alive flies; LT50 was calculated as the number of days it takes for 50% of the flies to die. For infection, LB medium (3ml) was inoculated from glycerol stocks of bacteria strains stored at −80 °C and cultured overnight at 37 °C with shaking at 200 rpm. The overnight cultures were diluted (1:100) and cultured in 3 mL LB until they reached an OD_600nm_: 3.0 (~3 × 10^9^ CFUs/mL). Anaerobic strains were grown in BHI medium (Brain Heart Infusion) (3 mL), a medium conducive to their growth, for 2 days at 37 °C without shaking in 5% O_2_ jars [82]; 0.5 mL of bacterial culture, equivalent to OD_600nm_: 3.0, was collected from each strain, centrifuged for 2 min at 6010 rcf; and the pellet (s) of the one or both strains was resuspended in 5 mL infection mix (1 mL 20% Sucrose, 3.5 mL ddH_2_O and 0.5 mL LB). The resulting 4% sucrose in the infection mix is the minimum concentration required for fly survival beyond 20 days at room temperature. Figure 2 indicates alternative infection mixes containing 80% LB (*), 95% LB (**), or 80% BHI (***). An autoclaved cotton ball was inserted in a narrow plastic fly vial, soaked with the infection mix, and plugged with a dry cotton ball. For fly infection under hypoxia, vials were covered with a piece of cotton net kept in place with rubber bands and placed in 5% O_2_ jars. Vials were transferred to 29 °C in all cases. Three vials containing 10 flies each were used for each infection experiment.

#### 4.3.2. Fly Intestinal Colonization Assay

Oregon R female flies 4–7 days old, previously starved for 5–6 h, were fed with the above-described infection mix for 1, 2, or 5 days. At each of the corresponding time points, three groups of flies, containing three flies each, were externally sterilized and homogenized using the Qiagen Tissuelyser LT for 5 min at 50 Hz, and dilutions were plated on LB plates for most strains and incubated overnight at 37 °C, or on BHI plates for sensitive-to-oxygen strains and incubated under anoxia for 2 days at 37 °C. Then, CFUs were enumerated for each triplicate of plates, and the number of CFUs per fly was calculated.

#### 4.3.3. Mitotic Cells Quantification

The quantification of mitosis in whole midgut tissues was performed by pH3 labeling, as described elsewhere [14]. Briefly, female adult flies were dissected in 1X PBS. Midguts were fixed in 1X PBS with 4% paraformaldehyde for 30 min at room temperature. Samples were washed in PBS 3 times for 10 min each. Then, the tissues were permeabilized in PBS with 0.1% Triton X-100 (for 20 min at room temperature) and blocked in a blocking solution containing PBS, Triton X-100 0.1%, 2.5% BSA, and 10% normal goat serum for 30 min at room temperature. All samples were incubated overnight with primary antibody mouse anti-pH3 (EMD Millipore Corp, St. Louis, MO, USA) at a dilution of 1:4000. After washing 3 times for 10 min each time with washing solution (PBS, 0.1% Triton X-100) and horizontal shaking, samples were incubated with secondary goat anti-mouse antibody conjugated with Alexa Fluor 488 or 546 (Thermo Fisher Scientific, Waltham, MA, USA) for 1–2 h at room temperature at a dilution of 1:2000. DNA was visualized with DAPI (1 mg/mL, Sigma), diluted 1:3000. Representative images were acquired using a Leica SP2 confocal laser-scanning inverted microscope with a 20X objective lens.

#### 4.3.4. Assessing PA14 Growth—CFUs (Colony-Forming Units) Assay

Freshly LB agar-plated bacterial single colonies from *P. aeruginosa* (PA14), *S. marcescens, Klebsiella (aerogenes* and sp.), *A. baumannii, S. paratyphi*, and *Providencia* sp. were inoculated in LB medium (3 mL) in the absence or presence of 4% glucose and cultured overnight (~16 h), at 37 °C, with shaking (200 rpm). Bacteria cultures were centrifuged (5 min at 6010 rcf) and bacterial supernatants isolated from pellets. Bacterial supernatants were filtered (using 0.2 μm filters) to remove any bacteria cell residues, whereas bacterial pellets were resuspended in 1.5 mL fresh LB and then immediately heat-killed (70 °C for 7–10 min). Filtered bacterial supernatants or heat-killed bacteria were mixed with PA14 bacteria (30 μL) in fresh LB (1.5 mL) (1:1 volume ratio), in a glass test tube, and incubated for 24 h at 37 °C with shaking at 200 rpm. The bacteria suspension mix (100 μL) was streaked in selective LB agar plates containing the antibiotic rifampicin (50 μg/mL). PA14 colonies were enumerated after overnight incubation at 37 °C.

#### 4.3.5. PA14 Pyocyanin Measurement

Freshly LB agar-plated bacterial single colonies from *P. aeruginosa* (PA14), *S. marcescens, Klebsiella (aerogenes* and sp.), *A. baumannii, S. paratyphi*, and *Providencia* sp. were inoculated in LB medium (3 mL) in the absence or presence of 4% glucose and cultured overnight (~16 h), at 37 °C, with shaking (200 rpm). Bacteria cultures were measured with the spectrophotometer (OD_600 nm_) and diluted appropriately in order to have the same spectrophotometric measurements. Bacteria cultures were centrifuged (5 min at 6010 rcf) and bacterial supernatants isolated from pellets. Bacterial supernatants were filtered (using 0.2 μm filters) to remove any bacteria cell residues, whereas bacterial pellets were resuspended in 15 mL fresh LB and then immediately heat-killed (70 °C for 10–18 min). Filtered bacterial supernatants or heat-killed bacteria were mixed with PA14 bacteria (300 μL) in fresh LB (15 mL) (1:1 volume ratio), in a glass test tube, and incubated for 24 h at 37 °C with shaking at 200 rpm. Cultures were centrifuged (10 min at 3381 rcf) and supernatant collected. Following the addition of chloroform (4.5 mL to 7.5 mL of supernatant), vortexing, and centrifugation (10 min at 3381 rcf), a blue layer formed, 3 mL of which was transferred to a new tube. HCl (1.5 mL, 0.2 M) was added, samples were vortexed (2 times for 10 s) and centrifuged (2 min at 3381 rcf), and 1 mL of the pink layer was transferred to cuvettes for pyocyanin measurement. The pyocyanin concentration (μg/mL) was calculated by multiplying the spectrophotometric measurements taken at OD_520 nm_ by 17.07 and then multiplying them again by 1.5 due to chloroform dilution.

### 4.4. Mice Diet, Maintenance and Experiments

Five-week-old female CD1 mice were obtained from ‘The Cyprus Institute of Neurology and Genetics’ (CING) and reared (5 per cage) at 24 °C in a 12-h day and night cycle. Mice were adapted to their new environment for 3 days before being used for experiments. A standard chow diet (a complete balanced diet containing mainly starch 35.18%, sucrose 5.66%, crude protein 22%, and crude oil 3.5%) was obtained from Mucedola s.r.l Italy.

#### Intranasal Mouse Lung Infection Assay

The intranasal infection achieves the spreading of the bacteria from the upper airways to the intestine and low airways, thus mimicking the pathology seen in acute bacterial pneumonia [83,84]. Bacterial strains used were PA14, *E. coli* MGH, *Klebsiella (aerogenes* and sp.), *A. baumannii, Providencia* sp., and lactic acid bacteria *L. plantarum* and *B. infantis.* PA14 was grown in LB liquid cultures overnight (~16 h) and then diluted 1:100 and grown over day for about 5 h in LB to OD_600_ nm: 3.0 (~3 × 10^9^ CFUs/mL). *E. coli* MGH, *Klebsiella (aerogenes* and sp.), *A. baumannii*, and *Providencia* sp. were grown in LB liquid cultures overnight (~16 h) and then diluted 1:100 and grown over day in 4% Glucose (LB) to OD_600 nm_: 3.0 (~3 × 10^9^ CFUs/mL). *L. plantarum* and *B. infantis* were grown for 48 h under hypoxic conditions (5% O_2_) in 4% Glucose (LB). Each culture (1 mL) was centrifuged (3 min at 6010 rcf) and the supernatant isolated and diluted directly in 4% Sucrose (LB). Mice were intranasally infected under very light anaesthesia (100 μL solution containing anaesthetics Ketamine—100 mg and Xylazine—20 mg), as previously described [67,85], by placing 10 μL of a bacterial suspension in each nostril (20 μL in total) to reach the desired infectious dose of 2 × 10^7^ CFUs per mouse. Mortality counts were taken twice a day for a week.

### 4.5. Computational Analysis

Pairwise comparisons of bacterial CFUs and pyocyanin production were evaluated using the Mann–Whitney U-test and one-way analysis of variance (ANOVA) with post-hoc Dunn’s multiple comparison test for samples < 10. The survival curves of the flies and mice were analyzed with the Kaplan–Meyer method and the log-rank test.

### 4.6. Ethics Issues

The animal protocols were approved by the Cyprus Veterinary Service inspectors under the license number CY/EXP/PR.L6/2018 for the Laboratory of Prof. Apidianakis at the University of Cyprus. The veterinary services act under the auspices of the Ministry of Agriculture in Cyprus, and the project number is CY.EXP101. These national services abide by the National Law for Animal Welfare of 1994 and 2013 and the Law for Experiments with Animals of 2013 and 2017. All experiments were performed in accordance with these guidelines and regulations.

## Figures and Tables

**Figure 1 metabolites-12-00449-f001:**
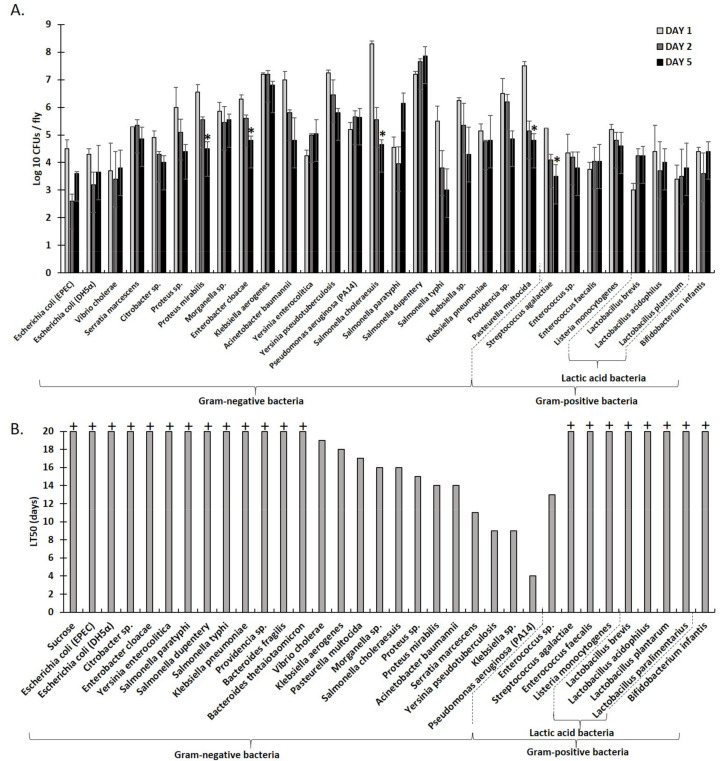
Culturable strains spanning a spectrum of human-associated bacterial species are retrievable from and may kill *Drosophila* upon feeding infection. Colony-forming units (CFUs) at Day 1, Day 2, and Day 5 of the feeding of female *Drosophila melanogaster* Oregon R flies (**A**), with each of the 22 different Gram negative bacteria strains or each of the 8 different Gram positive bacteria strains, 4 of which were lactic-acid bacteria. A statistical analysis was performed to compare log10 CFUs/fly between Days 1, 2, and 5 for each bacterial strain using one-way ANOVA. Significant differences at *p* < 0.05 were only found between Days 1 and 5 for five strains, indicated with *. Data from six replicates using 30 flies per replicate are plotted. (**B**) The survival of female *Drosophila* evaluated as the number of days it takes to reach 50% fly mortality (LT50) following feeding infection with each of the 24 different Gram negative bacteria strains or each of the 9 different Gram positive bacteria strains, 5 of which were lactic-acid bacteria. The plus sign (+) over the bar indicates that LT50 was not reached by day 20. Data from 2 replicates using 30 flies per replicate are plotted.

**Figure 2 metabolites-12-00449-f002:**
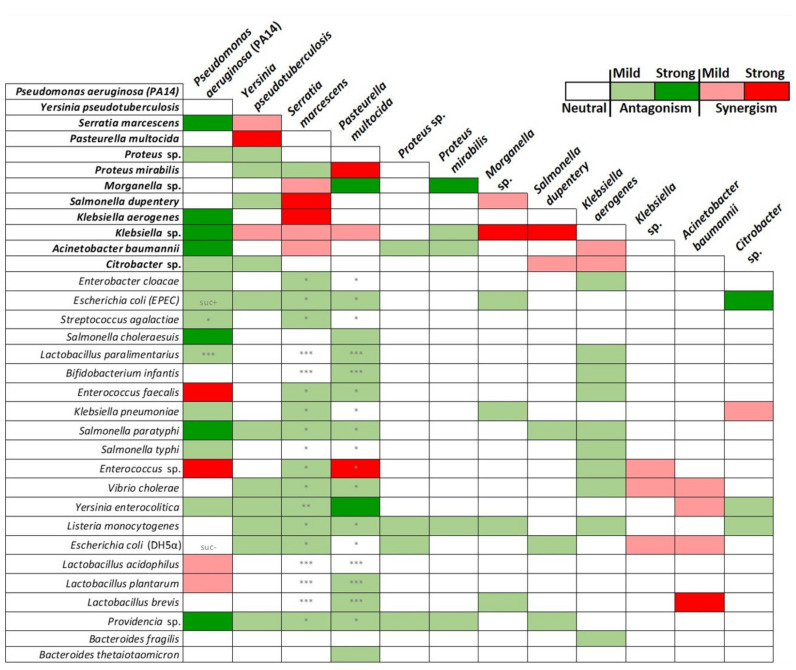
The classification of bacterial interactions as neutral, synergistic, or antagonistic, based on *Drosophila* survival upon feeding infection. *Drosophila* survival, as days required to reach 50% fly mortality (LT50), following feeding infection with 12 virulent human pathogenic bacteria strains (indicated in bold and at the top of each row—LT50_H_) in combination with 33 other bacteria strains (occupying a horizontal row each—LT50_H+X_). Neutral interactions (white boxes) did not exhibit an effect on fly survival imposed by the most virulent strain. Synergistic interactions exhibited enhanced lethality, either mild (LT50_H_/LT50_H+X_ = 1.2–1.5; indicated by pink boxes) or strong (LT50_H_/LT50_H+X_ > 1.5; indicated by red boxes). Antagonistic interactions exhibited reduced lethality, either mild (LT50_H_/LT50_H+X_ = 0.5–0.8; indicated by light green boxes) or strong (LT50_H_/LT50_H+X_ < 0,5; indicated by dark green boxes). Infection media contained 10% LB, except in the cases indicated with asterisks: * contained 80% LB, ** contained 95% LB, and *** contained 80% BHI. suc +/− indicate *E. coli* strains positive/negative for being able to metabolize sucrose. Data were collected from a minimum of 30 flies per condition per infection.

**Figure 3 metabolites-12-00449-f003:**
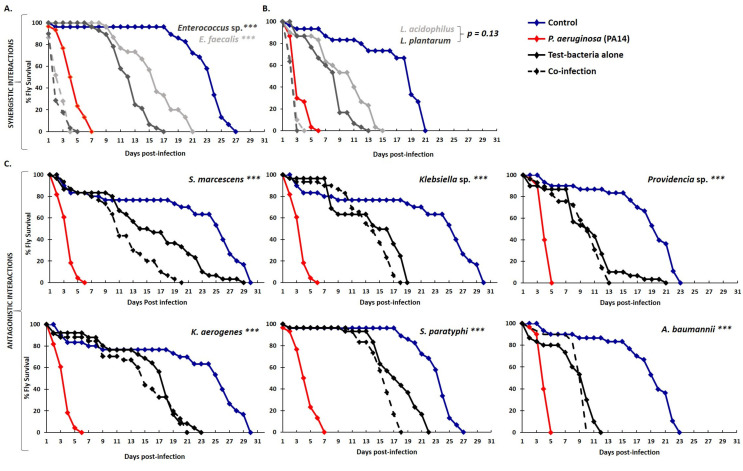
*P. aeruginosa* displays synergistic interactions with *Enterococci* (and tentatively so with lactic acid bacteria) and antagonistic interactions with Proteobacteria strains upon *Drosophila* feeding infection. (**A**–**C**) Kaplan–Meyer survival curves of flies feeding on 4% sucrose (blue lines) or infected with either *P. aeruginosa* (PA14) only (red lines) or *E. faecalis* only (solid light grey line, in A) or *Enterococcus* sp. only (solid dark grey line, in A) or co-infected PA14 + *E. faecalis* (dotted light grey line, in (**A**)) or PA14 + *Enterococcus* sp. (dotted dark grey line, in (**A**)). The statistical significance was observed for both co-infections vs. PA14 (*** *p* < 0.0001). (**B**) *L. acidophilus* only (solid light grey line) or *L. plantarum* only (solid dark grey line) or co-infection PA14 + *L. acidophilus* (dotted light grey line) or PA14 + *L. plantarum* (dotted dark grey line). The statistical significance of co-infections vs. PA14 was *p* = 0.132 and *p* = 0.131, respectively. (**C**) The test bacteria alone, namely, *S. marcescens*, *Klebsiella* sp., *Providencia* sp., *K. aerogenes*, *S. paratyphi*, or *A. baumannii* (solid black line), or co-infection with PA14 (dotted black line). Statistical significance was observed for all co-infections vs. PA14 (*** *p* < 0.0001). Data were collected from two independent experiments, with a minimum of 30 flies per experiment.

**Figure 4 metabolites-12-00449-f004:**
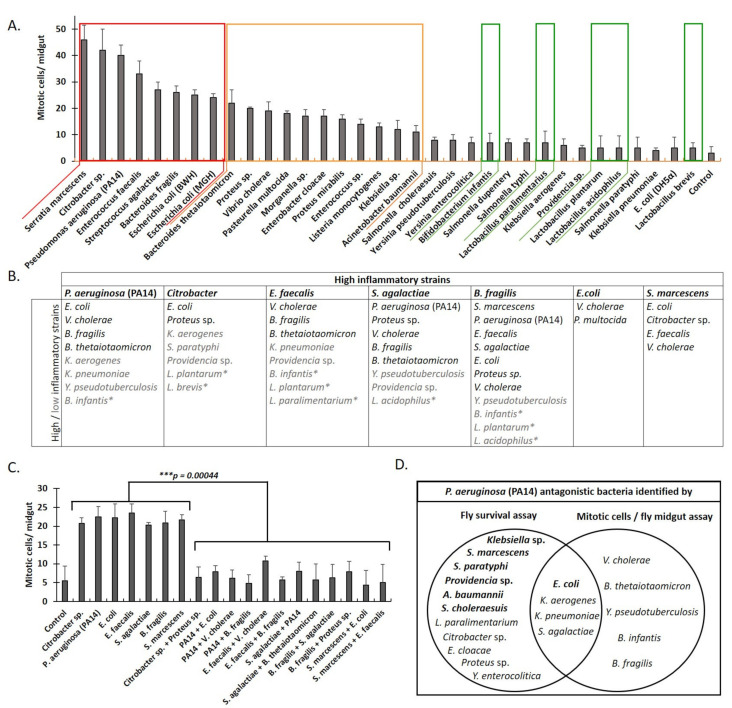
The classification of bacterial strains as virulent or antagonistic to virulent strains by assessing midgut cell mitosis upon fly feeding. Mitosis (phospho-Histone 3 positive cells per midgut) in flies fed with a single bacterial strain (**A**,**C**) or in combination with an antagonistic strain (**B**,**C**). (**A**) Red, orange, and green rectangles indicate increased, moderate, or minimal mitosis, respectively, upon infection. Data from two independent experiments assessing 30 flies in total. (**B**) Antagonistic interactions between highly regenerative bacteria strains (indicated in bold) and other highly (black) or lowly (grey) regenerative bacteria strains, including lactic acid bacteria (indicated by *), leading to >50% reduction in mitosis. (**C**) Statistically significant differences between single virulent strains shown vs. the corresponding co-infections at *** *p* < 0.0005. (**D**) Fly survival vs. midgut mitosis assay overlap of bacterial strains exhibiting antagonism against *P. aeruginosa*. Bacteria strains indicated in bold displayed stronger antagonistic interactions than the rest.

**Figure 5 metabolites-12-00449-f005:**
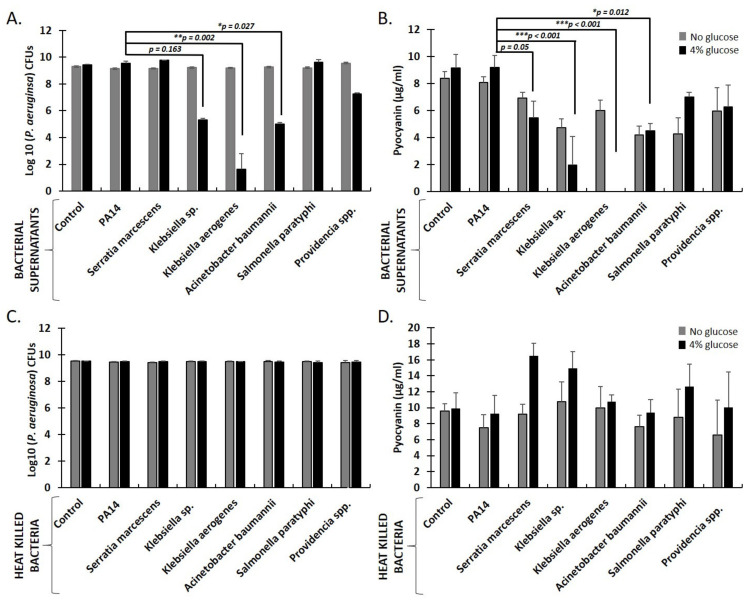
The secreted factors from the Proteobacteria, *K. aerogenes, Klebsiella* sp., and *A. baumannii* but not the corresponding heat-killed bacteria inhibit *P. aeruginosa* (PA14) growth and pyocyanin production in culture, in the presence of glucose under normoxia. CFUs of *P. aeruginosa* (PA14) growth (**A**) and pyocyanin concentration (**B**) in cultures under normoxia, in the absence or presence of 4% glucose, at 24 h, of PA14 alone or in the presence of bacterial supernatants derived from cultures grown under normoxia, of the species *S. marcescens*, *K. aerogenes*, *Klebsiella* sp., *A. baumannii*, *S. paratyphi*, and *Providencia* sp.. CFUs of *P. aeruginosa* (PA14) growth (**C**) and pyocyanin concentration (**D**) in cultures under normoxia, in the absence or presence of 4% glucose, at 24 h, of PA14 alone or in the presence of heat-killed bacteria of the species *S. marcescens*, *K. aerogenes*, *Klebsiella* sp., *A. baumannii*, *S. paratyphi*, and *Providencia* sp. Error bars represent the standard deviation of the mean. A statistical analysis was performed using the one-way ANOVA test and Dunn’s multiple comparisons test. Statistical significance is indicated as n/s for *p* > 0.05, * *p* < 0.05, ** *p* < 0.005, and *** *p* < 0.0005.

**Figure 6 metabolites-12-00449-f006:**
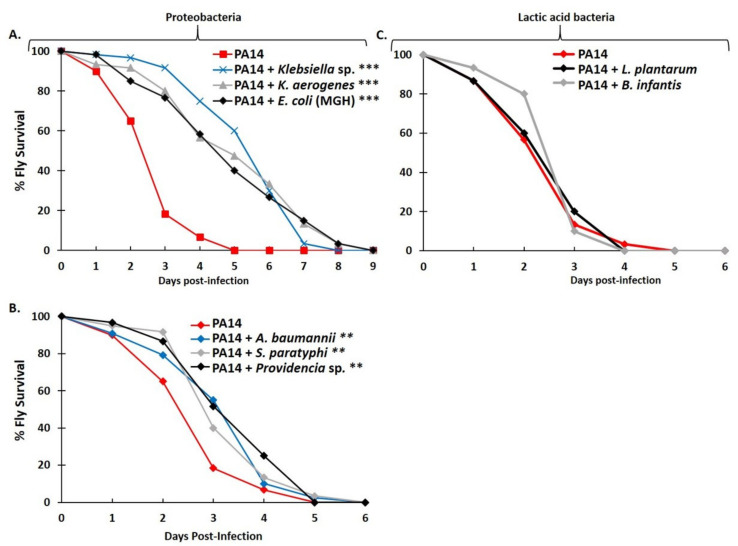
*Drosophila* feeding infection and survival under mildly hypoxic conditions ([O_2_] = 5.5%) are conducive to antagonistic interactions between *P. aeruginosa* (PA14) and certain Proteobacteria but not lactic acid bacteria. (**A**) The survival curves of flies infected with *P. aeruginosa* (PA14) only (red lines) or co-infected with either PA14 and *Klebsiella* sp. (blue line) or PA14 and *K. aerogenes* (grey line) or PA14 and *E. coli* (MGH) (black line). The statistical significance in all cases at *** *p* < 0.0001. (**B**) The survival curves of flies infected with *P. aeruginosa* (PA14) only (red lines) or co-infected with PA14 and *A. baumannii* (dark blue line, *p* = 0.004) or *Providencia* sp. (light blue line, *p* = 0.001) or *S. paratyphi* (grey line, *p* = 0.001). The statistical significance in all cases at ** *p* < 0.004. (**C**) The survival curves of flies infected with *P. aeruginosa* (PA14) only (red lines) or co-infected with either PA14 and *L. plantarum* (grey line) or PA14 and *B. infantis* (black line). No statistical significance was observed. Data were collected from two independent experiments, with a minimum of 30 flies per condition, per experiment.

**Figure 7 metabolites-12-00449-f007:**
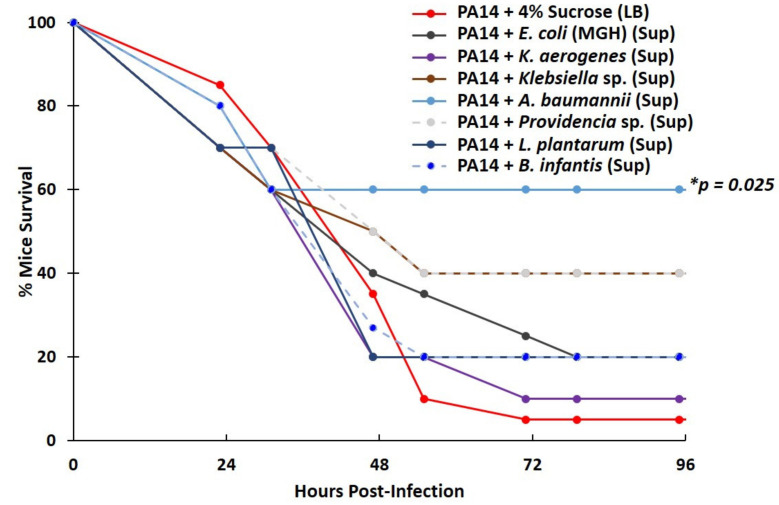
The supernatants derived from lactic acid bacteria, grown under anoxia, failed to inhibit *P. aeruginosa* (PA14) virulence in a mouse lung-infection model. The survival curves of CD1 mice, following intranasal infection with *P. aeruginosa* (PA14) using 4% sucrose in LB as a vehicle alone or 4% sucrose in LB supernatants of Proteobacteria strains (*K. aerogenes, Klebsiella* sp., *A. baumannii*, and *Providencia* sp.) or lactic acid bacteria (*L. plantarum, B. infantis*); the latter were grown under severe hypoxia conditions (static cultures at an ambiance of [O_2_] = 5%). A statistical analysis of survival differences between the group infected with PA14 (*n* = 20) vs. each of the groups infected with PA14 delivered in the supernatant from either *A. baumannii* (* *p* = 0.025; *n* = 10), *Providencia* sp. (*p* = 0.123; *n* = 10), *Klebsiella sp.* (*p* = 0.166; *n* = 10), *K. aerogenes* (*p* = 0.826; *n* = 10), *L. plantarum* (*p* = 0.796; *n* = 10), or *B. infantis* (*p* = 0.809; *n* = 10).

**Figure 8 metabolites-12-00449-f008:**
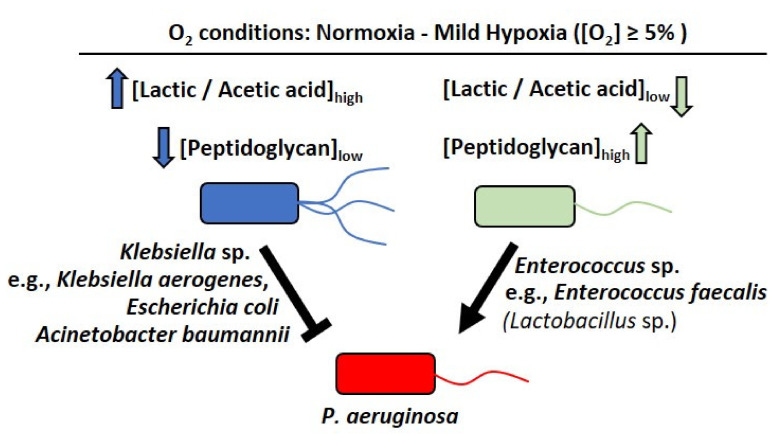
A schematic representation of bacterial interactions involving *P. aeruginosa*, under normoxia to mild hypoxia ([O_2_] ≥ 5%). Under normoxia to mild hypoxia (O_2_ ≥ 5%) encountered in the human gut mucosa and the lung epithelium, aerotolerant Firmicutes species, such as *Enterococcus* sp. (e.g., *E. faecalis*) and lactic acid bacteria (e.g., *Lactobacillus* sp.), do not favor anaerobic respiration ([lactic/acetic acid]_low_) but shed [peptidoglycan]_high_ (per Korgaonkar et al., 2011), which is sensed by *P. aeruginosa*, leading to synergistic virulence [31]. Under the same conditions, Proteobacteria, specifically *Klebsiella* sp. (e.g., *K. aerogenes*), *A. baumannii*, and *E. coli*, undergo aerobic fermentation ([lactic/acetic acid]_high_) in the presence of glucose, which, in combination with their low production of peptidoglycan ([peptidoglycan]_low_), facilitates antagonistic interactions with *P. aeruginosa*.

## Data Availability

Data is contained within the article or Appendix A.

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
