# Peer review of "Proteobacteria and Firmicutes Secreted Factors Exert Distinct Effects on Pseudomonas aeruginosa Infection under Normoxia or Mild Hypoxia"

_metabolites, 2022, doi:10.3390/metabo12050449_

Round 1

Reviewer 1 Report

The work reports a Drosophila-based assay study the impact of human bacterial interactions in terms of fly survival and midgut regeneration in a O2 dependent manner. Collectively the manuscript is innovative and holds translational potential. The authors are requested to make following modifications:

  • The applicability of the study model to understand host-microbiota and microbe-microbe interaction in relation to human disease needs more emphasis.
  • Fig 1A, no statistical tool has been used to compare day1, 2 and 5.
  • Fig 2, y-axis is inconsistent
  • Mention whether the animal diet was endotoxin-free.

Author Response

Comment 1: The applicability of the study model to understand host-microbiota and microbe-microbe interaction in relation to human disease needs more emphasis.

Response: The reviewer is right to request greater emphasis to be given to the applicability of the study model to understand host-microbiota and microbe-microbe interaction, in relation to human disease. This has now been emphasized in the Discussion section (see lines 443-449 and 471-474).

Comment 2: Fig 1A, no statistical tool has been used to compare day1, 2 and 5.

Response: We have now performed statistical analysis using one-way ANOVA to compare log 10 CFUs/ fly between days 1, 2 and 5 for each individual bacteria strain. No statistical significance was observed when comparing log 10 CFUs/ fly between days 1 and 2 or days 2 and 5. Statistical significance was observed however in some cases when comparing log 10 CFUs/ fly between days 1 and 5 and is now indicated with * in Figure 1A that corresponds to a p value < 0.05. It is also mentioned in the text (lines 122-125) and in the Figure legend (lines 150-153).

Comment 3: Fig 2, y-axis is inconsistent

Response: In Figure 2 the y axis represents percentage fly survival and it is consistent. We believe that the reviewer here is referring to the x-axis being inconsistent between graphs. This has now been fixed (x-axis of all graphs now begins at Day 1 and continue up to and including Day 31).

Comment 4: Mention whether the animal diet was endotoxin-free

Response: The animal diet was endotoxin-free. This is now mentioned in Materials and methods (see line 581-582). 

Reviewer 2 Report

The article Proteobacteria and Firmicutes secreted factors exert distinct effects on Pseudomonas aeruginosa infection under normoxia or mild hypoxia  brings interesting results with outcomes for further pre-clinical investigation. However I recommend to read whole article again and use correct names of bacteria (line 19,11, 237-240, etc) and check more precisely text.

Author Response

Comment 1: The article Proteobacteria and Firmicutes secreted factors exert distinct effects on Pseudomonas aeruginosa infection under normoxia or mild hypoxia  brings interesting results with outcomes for further pre-clinical investigation. However I recommend to read whole article again and use correct names of bacteria (line 19,11, 237-240, etc) and check more precisely text.

Response: We believe the reviewer is referring to lack of italicization regarding bacteria names. This has now been corrected throughout the text.

Reviewer 3 Report

The aim of the study was to investigate a very interesting issue of microbes coexistence under normoxia and hypoxia conditions. It is obviously seen that a lot of effort went into the study. Novelty and some findings made on this basis, in my opinion, are very interesting, especially for a particular group of readers. However, the manuscript needs some profound re-writing.

My major concerns are:

  • The number of mistakes in the nomenclature/names of bacteria is inacceptable for the manuscript of microbiological origin, e.g. Escherischia, Providentia, paratyphi, Cholerae, agalactaea, clocae, etc. plus, full name of bacteria should be used only once in the body of manuscript – first time used only.
  • What about control performed for the experiments shown on Figure 1A?
  • What does Enterococcus means? You applied mixture of species or did not identify the isolate to the genus level?
  • It should be explained why lactic acid bacteria have so limited effect on epithelial cells turnover.
  • It should be discussed why co-infection with particular strains influenced fly survival so strongly.
  • Line 449 – what were the criteria to assign particular strains/bacteria to highly virulent strains?
  • It should be described what the purpose of rifampicin addition to media was.
  • A number of sentences from Results section should be deleted or moved to Introduction, M&M or (better) Discussion because they repeat or do not present results from the study itself.

Minor concerns are:

  • The last part of Introduction is not an Introduction.
  • Keywords should be listed in alphabetical order, in my opinion.
  • Line 59 – References should be combined.
  • Sentences should not start with a number.
  • It should be stated why this particular reagents concentration was applied for the study purpose.
  • There are some repeats in the text/tables/figures that should be deleted before the acceptance of the manuscript for publication.
  • Italics missing in a number of places and some typing errors should be corrected before the publication.

All the points mentioned above do not decrease the overall value of the research.

Author Response

The aim of the study was to investigate a very interesting issue of microbes coexistence under normoxia and hypoxia conditions. It is obviously seen that a lot of effort went into the study. Novelty and some findings made on this basis, in my opinion, are very interesting, especially for a particular group of readers. However, the manuscript needs some profound re-writing. 

My major concerns are:

  • The number of mistakes in the nomenclature/names of bacteria is inacceptable for the manuscript of microbiological origin, e.g. Escherischia, Providentia, paratyphi, Cholerae, agalactaea, clocae, etc. plus, full name of bacteria should be used only once in the body of manuscript – first time used only.

Bacteria names have been corrected in terms of spelling and italicization- full names are now only used the first time of reference and then abbreviated appropriately both in the text and figures.

  • What about control performed for the experiments shown on Figure 1A?

The sucrose control (uninfected flies) has now been included in the graph of Figure 1A. The reviewer is correct to consider microbiota and natural flora, however and as now clarified in section 4.3 (line 582-585), for infection experiments, flies were transferred to clean vials with fresh fly food every day, containing preservatives (propionic acid and Tegosept), which helped to eliminate microbiota and to avoid pre-treatment with antibiotics that would interfere with subsequent bacterial infection and colonization. As a result, no colonies were detected on LB or BHI plates, hence log10 CFUs/ fly = 0, in the sucrose control.

  • What does Enterococcusmeans? You applied mixture of species or did not identify the isolate to the genus level?

instead of spp. is correctly used in the case of Enterococcus and other strains, because we did not use a mixture of Enterococcus species or a mixture of any genus species. Where “sp.” is indicated, we mean that the strain is identified at the genus level and not identified at the species level.

  • It should be explained why lactic acid bacteria have so limited effect on epithelial cells turnover.

The reviewer is correct to notice that lactic acid bacteria have a limited effect on epithelial cell turnover. We suggest that Lactobacillus strain differences or experimental set up may account for the low mitosis induction in our study. We now provide such an explanation in Discussion section (lines 487-493).

  • It should be discussed why co-infection with particular strains influenced fly survival so strongly.

We believe the reviewer here is referring to the mechanism of action used by antagonistic bacteria, to exert such a strong positive influence on fly survival. In this work, we did not aim to prove the mechanism, but in accordance with our previous findings (https://pubmed.ncbi.nlm.nih.gov/31595010/) showing that simple sugars (glucose or sucrose) are essential for E. coli strains to produce lactic and acetic acid and inhibit P. aeruginosa growth, we now show that (i) flies infected via feeding with P. aeruginosa in combination with Klebsiella sp., Klebsiella aerogenes or E. coli (MGH), in the presence of sucrose, exhibited a significant delay in the timing of fly mortality (Figure 5A) and (ii) glucose is necessary to facilitate inhibition of P. aeruginosa growth by the supernatants of Proteobacteria strains in liquid cultures (Figure 4A left). Therefore, as now mentioned in section 2.4, discussed in the Discussion section (lines 521-525) and proposed in Figure 7, we believe that Proteobacteria in general, must use sugar fermentation, in combination with their low production of peptidoglycan, to antagonize P. aeruginosa virulence.  

  • Line 449 – what were the criteria to assign particular strains/bacteria to highly virulent strains?

Particular strains were designate as highly virulent, based on preliminary fly survival and midgut mitosis comparative assessments, similar to the ones shown in Figures 1B and 3A respectively. Following this initial categorization, upon co-infection, some highly virulent strains that were close to the cut-off may have proven less virulent than the bacterial strain with which they were co-infected.  

  • It should be described what the purpose of rifampicin addition to media was.

Rifampicin was added only in the case of test-tube culture experiments (not in any of the fly or mice experiments), to ensure that no other bacteria would grow on LB agar plates for the purpose of measuring CFUs of P. aeruginosa strain PA14 mixed with other bacterial strains. This has now been clarified in Materials and Methods (lines 559-560).  

  • A number of sentences from Results section should be deleted or moved to Introduction, M&M or (better) Discussion because they repeat or do not present results from the study itself.

A number of sentences, offering interpretation of results were deleted from the Results section and moved to the Discussion section as suggested by the reviewer.

Minor concerns are:

  • The last part of Introduction is not an Introduction.

The reviewer is correct to note that the last paragraph of the introduction is not an introduction but a summary of this work’s findings. We consider this a common practice in research articles that can prove helpful to readers. However, we have edited this paragraph to limit its extent.   

  • Keywords should be listed in alphabetical order, in my opinion.

Keywords are now listed in alphabetical order.

  • Line 59 – References should be combined.

References are now combined.

  • Sentences should not start with a number.

This has now been corrected – wherever a sentence started with a number, this has now been replaced by a word.

  • It should be stated why this particular reagents concentration was applied for the study purpose.

We believe the reviewer here is referring to the fact that infection media was supplemented with 10% LB. The original protocol for infection of Drosophila via feeding was established in 2009 by the corresponding author (https://www.pnas.org/doi/10.1073/pnas.0911797106). This protocol was extensively revised and through trial and error we have now established that 10% LB is necessary for reproducibility of results, along with the use of female flies and a rearing temperature of 29oC, as now described in section 2.1 (lines 106-108).

  • There are some repeats in the text/tables/figures that should be deleted before the acceptance of the manuscript for publication.

We have gone over the text/tables/figures and have deleted any repetitions detected.

  • Italics missing in a number of places and some typing errors should be corrected before the publication.

Italics have been introduced and typing errors corrected.

All the points mentioned above do not decrease the overall value of the research.

Reviewer 4 Report

The manuscript is well written and shows interesting data on the effect of various bacteria on Pseudomonas aeruginosa virulence. The manuscript is acceptable for publication after doing some minor corrections as stated below.

Minor comments:

Line 59: The references should be within the same brackets.

Figure 2: In the legend at the upper right side: I think that the wording "Antagonist" is not right word. Did you intend the "test bacteria alone"?

In Table 1: All bacteria except for E.coli were written in full name. To keep with the consistency, please also write the E. coli in full name.

Figure 2: In the panel of K. aerogenes the black solid line lacks symbols. Please correct.

In lines 234-240: The bacteria names should be in italics. The same for Section 4.2.

In the text: While most bacteria are described in abbreviations, Enterococcus faecalis appears in full name. Choose either of the options.

In Figure 7: correct to: e.g.,

Line 546: Correct to strains.

Line 596 and each time X-100 appears: Correct to Triton X-100.

Ethic issues need to be added to the Method Section.

Author Response

Comment 1: Line 59: The references should be within the same brackets.

Response: The references have now been placed within the same brackets.

Comment 2: Figure 2: In the legend at the upper right side: I think that the wording "Antagonist" is not right word. Did you intend the "test bacteria alone"?

Response: The reviewer is correct – the word “Antagonist” in the legend of Figure 2 within the picture as well as within the text (see line 227) has now been replaced with “test bacteria alone”.

Comment 3: In Table 1: All bacteria except for E.coli were written in full name. To keep with the consistency, please also write the E. coli in full name.

Response: The reviewer is correct – E. coli has been replaced by Escherichia coli, to keep with the consistency of the rest of the bacteria names within the table.

Comment 4: Figure 2: In the panel of K. aerogenes the black solid line lacks symbols. Please correct.

Response: Symbols have been added to the black solid line representing K. aerogenes.

Comment 5: In lines 234-240: The bacteria names should be in italics. The same for Section 4.2.

Response: Bacteria names have been italicized in indicated lines and sections.

Comment 6: In the text: While most bacteria are described in abbreviations, Enterococcus faecalis appears in full name. Choose either of the options.

Response: The reviewer is correct - Enterococcus faecalis no longer appears in full name, but abbreviated for consistency.

Comment 7: In Figure 7: correct to: e.g.,

Response: In Figure 7, e.g. has been corrected to e.g.,

Comment 8: Line 546: Correct to strains.

Response: In previously Line 546 currently 575, stains has been corrected to strains.

Comment 9: Line 596 and each time X-100 appears: Correct to Triton X-100.

Response: X-100 has been corrected to Triton X-100.

Comment 10: Ethic issues need to be added to the Method Section.

Response: Ethics issues have been added as section 4.6. of the Methods section and removed from Notes (Institutional review Board statement) at the end of the paper.

Round 2

Reviewer 3 Report

There are still some mistakes in the nomenclature, e.g. Salmonella and full names of bateria species repeated in the manuscript.

For the reagent concentration, I ment the explanation of each reagent (or citation of some orginal research that applied that before), not LB only.

Author Response

We kindly thank the reviewer for the constructive comments. Below please find our responses. 

1. There are still some mistakes in the nomenclature, e.g. Salmonella and full names of bateria species repeated in the manuscript.

The reviewer is correct to notice that there are still some mistakes in the nomenclature. These have now been corrected. Full names are now only mentioned upon first appearance and then appropriate abbreviations are used thereafter.

2. For the reagent concentration, I meant the explanation of each reagent (or citation of some original research that applied that before), not LB only.

To clarify, the pertinent section (4.3.1. Fly infection and survival) is now extensively edited. Regarding reagent concentrations: Based on our experience, 4% sucrose, is the minimum concentration required for flies to survive beyond 20 days. This concentration does not hamper Pseudomonas aeruginosa virulence, while higher concentrations of sugars may severely interfere with acute bacterial virulence. Our protocol has been used in Apidianakis et al PNAS 2009, Tamamouna et al Development 2020 and Evangelou et al G3 2019 (PMC6829147). Moreover, an original research paper (PMC5783981) is now cited to justify the use of BHI for the growth of anaerobic bacteria.